# MicroRNA-Based Combinatorial Cancer Therapy: Effects of MicroRNAs on the Efficacy of Anti-Cancer Therapies

**DOI:** 10.3390/cells9010029

**Published:** 2019-12-20

**Authors:** Hyun Ah Seo, Sokviseth Moeng, Seokmin Sim, Hyo Jeong Kuh, Soo Young Choi, Jong Kook Park

**Affiliations:** 1Department of Biomedical Science and Research Institute for Bioscience & Biotechnology, Hallym University, Chunchon 24252, Korea; santab1225@naver.com (H.A.S.); sokvisethmoeng@yahoo.com (S.M.); sychoi@hallym.ac.kr (S.Y.C.); 2Generoath, Seachang-ro, Mapo-gu, Seoul 04168, Korea; seokmin_sim@generoath.com; 3Department of Medical Life Sciences, College of Medicine, The Catholic University of Korea, Seoul 06591, Korea; hkuh@catholic.ac.kr

**Keywords:** microRNA, cancer, therapeutic resistance, chemosensitization, combination therapy

## Abstract

The susceptibility of cancer cells to different types of treatments can be restricted by intrinsic and acquired therapeutic resistance, leading to the failure of cancer regression and remission. To overcome this problem, a combination therapy has been proposed as a fundamental strategy to improve therapeutic responses; however, resistance is still unavoidable. MicroRNA (miRNAs) are associated with cancer therapeutic resistance. The modulation of dysregulated miRNA levels through miRNA-based therapy comprising a replacement or inhibition approach has been proposed to sensitize cancer cells to other anti-cancer therapies. The combination of miRNA-based therapy with other anti-cancer therapies (miRNA-based combinatorial cancer therapy) is attractive, due to the ability of miRNAs to target multiple genes associated with the signaling pathways controlling therapeutic resistance. In this article, we present an overview of recent findings on the role of therapeutic resistance-related miRNAs in different types of cancer. We review the feasibility of utilizing dysregulated miRNAs in cancer cells and extracellular vesicles as potential candidates for miRNA-based combinatorial cancer therapy. We also discuss innate properties of miRNAs that need to be considered for more effective combinatorial cancer therapy.

## 1. Introduction

Although cancer cells may initially respond to treatment, not all cells are eliminated. This limited efficacy of cancer therapies can be due to several resistance mechanisms, ultimately leading to the recurrence of cancer and associated death. Biological factors underlying therapeutic resistance include the expression levels of drug transporters, which limit the cytoplasmic concentrations of therapeutic agents [1]. The efficient repair of damaged DNA in cancer cells also contributes to therapeutic resistance, especially for treatments aimed at damaging DNA. Besides, autophagy can act as a pro-survival mechanism by interrupting apoptosis induction in cancer cells, thereby restricting the efficacy of cancer treatments [2,3].

There are other factors responsible for cancer therapeutic resistance. Cancer stem cells (CSCs) are known to be resistant to cancer treatments due to several features, such as self-renewal potential, activation of the DNA damage response, and high levels of drug transporter [4]. Autophagy is also known to support the properties of CSCs [5,6]. Additionally, epithelial–mesenchymal transition (EMT) has been revealed to confer the ability to acquire CSC properties onto cancer cells, thereby contributing to therapeutic resistance [7]. Moreover, cell-to-cell communication via extracellular vesicles among different types of cells within the cancer microenvironment could affect the efficacy of cancer therapies by delivering miRNAs that regulate various signaling pathways connected to therapeutic resistance [8,9].

Combination therapies have been proposed to overcome therapeutic resistance via the combined inhibition of different mechanisms. For example, the combination of cobimetinib and pictilisib was reported to be beneficial for the treatment of colorectal cancer cells. However, resistance is unavoidable even after the combination treatment [10]. Similarly, the simultaneous inhibition of phosphoinositide 3-kinase (PI3K) and a mechanistic target of rapamycin kinase (mTOR) was reported to activate extracellular signal-regulated kinase (ERK), a pro-survival factor, in acute myeloid leukemia [11]. Therefore, it is still necessary to explore new combination strategies to defeat therapeutic resistance. An improved understanding of the cellular basis of cancer therapeutic resistance can further provide promising opportunities to design and develop novel cancer treatment strategies to manage cancers.

MicroRNAs (miRNAs) are widely recognized, small, regulatory RNAs modulating numerous intracellular signaling pathways in several diseases, including cancers. Based on the expression levels and intracellular functions of miRNAs, they could act as tumor-suppressive or oncogenic factors in cancer cells [12,13,14]. The abnormal expression of miRNAs is associated with therapeutic resistance in cancer, and the modulation of miRNA levels, through either the inhibition or replacement approach, has been proposed to sensitize cancer cells to other anti-cancer therapies. This combination of miRNA-based therapy with other anti-cancer therapies (hereinafter referred to as miRNA-based combinatorial cancer therapy) is attractive due to the ability of miRNAs to regulate multiple resistance-mediating pathways by targeting multiple genes. However, it is indispensable to experimentally investigate whether the suppression or replacement of an miRNA can enhance the efficacy of anti-cancer therapies by efficiently impeding signaling pathways associated with therapeutic resistance, since the functions of miRNAs are dependent on the type of cancer. This article aims to elaborate on the significance of miRNA-based combinatorial cancer therapy in several types of cancer. We mainly focus on recent studies, which assess the target-related functions of miRNAs in association with their effects on anti-cancer therapies. We also discuss the characteristic features of miRNAs that exert influence on the adequate efficacy of miRNA-based combinatorial cancer therapy.

## 2. The Role of MiRNAs in Drug Efflux/Influx and Drug Sensitivity

### 2.1. Drug Transporters and Therapeutic Resistance

The limited intracellular concentration of anti-cancer drugs has been implicated in therapeutic resistance in various cancers. Of particular importance is the role of ATP-binding cassette transporters (ABC transporters) in the regulation of intracellular drug levels and the development of therapeutic resistance to multiple agents. ABC transporters are classified into seven subgroups, and the enhanced expression of several ABC transporters has been evaluated in cancer [1]. ABC transporters also contribute to the therapeutic resistance of CSCs. For instance, ATP-binding cassette subfamily C member 1 (ABCC1, also known as multidrug resistance protein 1, MRP1) and ABCB1 (also called multidrug resistance protein 1 (MDR1) and P-glycoprotein (P-gp)) are highly expressed in CSCs of several types of cancer, such as glioblastoma and breast cancer, and both ABCC1 and ABCB1 mediate the efflux of a number of therapeutic compounds [15]. In addition, although it is necessary to elucidate the precise function of solute carrier (SLC) transporters, they could regulate the cellular uptake of nutrients and chemotherapeutic agents. Therefore, the efficacy of some anti-cancer agents is associated with the expression pattern of SLC transporters [16].

### 2.2. MiRNAs Directly Targeting Drug Transporter Genes

Several miRNAs have been explored for their direct regulation of efflux and influx transporter levels to modulate the efficacy of therapeutic agents in cancer.

#### 2.2.1. ABCB1

The relationship between ABCB1 (MDR1, P-gp) and several miRNAs has been explored in cancer, showing the following: miR-129-5p reverses cisplatin resistance in gastric cancer cells by directly regulating ABCB1 [17]; miR-223-3p targets ABCB1 and regulates ABCB1-mediated multidrug resistance [18]; miR-491-3p, which targets ABCB1, improves the effectiveness of doxorubicin and vinblastine against hepatocellular carcinoma cells [19]; miR-495-3p increases the combinatorial effects of doxorubicin with paclitaxel on-cell growth and apoptosis in drug-resistant ovarian and gastric cancer cells via regulation of ABCB1 [20]; miR-508-5p targets ABCB1 and sensitizes cancer cells to several chemotherapeutic agents [21] (Table 1). Since miR-508-5p is a downstream gene of wild-type p53 [22], and p53 is the most frequently mutated in human cancer, the miR-508-5p-ABCB1 axis may be one of the possible reasons why ABCB1 is frequently over-expressed in cancer.

#### 2.2.2. ABCC1

In hepatocellular carcinoma cells, ABCC1 is targeted by both miR-133a-3p and miR-326. Over-expression of either miR-133a-3p or miR-326 successfully down-regulates ABCC1, hence sensitizing cells to doxorubicin [23]. It was also found that miR-1268a is one of the down-regulated miRNAs in temozolomide-resistant glioblastoma cells and directly modulates ABCC1 expression [24] (Table 1).

#### 2.2.3. ABCC5

ABCC5 (previously termed MRP5) is one of the determinants of gemcitabine cytotoxicity. Therefore, the modulation of ABCC5 expression can alter the sensitivity of cancer cells to gemcitabine [25]. There is consistent evidence that miR-210-3p reverses gemcitabine resistance by targeting ABCC5. Notably, it was unveiled that there is a negative correlation between ABCC5 and miR-210-3p expression levels in most cases of malignant pancreatic cancer tissues [26] (Table 1).

#### 2.2.4. ABCF2 and Other Transporters

ABCF2 is one of the members of ABCF transporters, a subgroup of the ABC superfamily. Although ABCF2 is unable to act as a membrane transporter owing to the absence of transmembrane domains, ABCF2 can mediate cisplatin resistance in cancer cells [27]. Further studies are required to examine how ABCF2 regulates cisplatin resistance. Nonetheless, it was indicated that miR-514 increases the efficacy of cisplatin by directly regulating ABCF2 [28]. Additional targets of miR-514 are ABCA1 and ABCA10, and the knockdown of either ABCA1, ABCA10, or ABCF2 can enhance the cytotoxicity of cisplatin [28]. In another study, it was pointed out that ABCG2 (also called breast cancer resistance protein 1, BCRP1) is a novel target of miR-328-3p and that the intracellular accumulation of mitoxantrone can be promoted by miR-328-3p [29] (Table 1).

#### 2.2.5. SLC19A1 and the Dual Role of a MiR-595

By contrast, SLC family 19 member 1 (SLC19A1), an influx transporter of methotrexate, was identified as a miR-595 target. As anticipated, the over-expression of miR-595 restricts the intracellular levels of methotrexate, as well as its cytotoxicity [30]. However, the function of miR-595 is the opposite in ovarian cancer cells. ABCB1 is another target of miR-595; therefore, over-expression of miR-595 could enhance the efficacy of cisplatin [31]. This indicates the possibility that miRNA has a dual role and regulates the sensitivity of cancer cells to anti-cancer agents differently, depending on the type of cancer (Table 1).

### 2.3. MiRNAs and the Transcription of Drug Transporter Genes

#### 2.3.1. ABCB1

There is a possibility that the expression of transporters is transcriptionally regulated in an miRNA-dependent way. Wnt/β-catenin is one of the signaling pathways regulating the transcription of ABCB1 and causes multidrug resistance in cancers [32]. Frizzled 7 (FZD7), a receptor for Wnt ligands, has been identified as an miR-27-3p target. Ectopic introduction of miR-27-3p down-regulates ABCB1 and augments 5-fluorouracil-induced cell death [33]. Recent studies also confirmed that miR-122-5p and miR-506-3p could elevate drug-induced cell death in hepatocellular carcinoma and colorectal cancer cells, respectively, by negatively regulating β-catenin and ABCB1 levels [34,35]. There is another possibility that miR-506-3p regulates ABCB1 expression by targeting the enhancer of zeste homolog 2 (EZH2), which is a possible transcription regulator of ABCB1 [36,37]. In addition, a decrease in ABCB1 expression can be mediated by miR-199a-3p and miR-218-5p via regulating mTOR and protein kinase C epsilon (PRKCE, also known as PKCε), respectively [38,39]. Since both mTOR and PKCε activate the signal transducer and activator of transcription 3 (STAT3), a transcription factor of ABCB1 [40,41,42], it is feasible that miR-199a-3p and miR-218-5p could down-regulate ABCB1 levels through a common mediator. As noted above, miR-491-3p directly controls ABCB1 expression. Interestingly, miR-491-3p also transcriptionally represses ABCB1 levels by regulating Sp3, which is known as a transcription factor of ABCB1 [19]. In addition, it was intriguingly noted that miR-508-5p, which directly regulates ABCB1, also targets zinc ribbon domain-containing 1 (ZNRD1), thereby negatively regulates the transcription of ABCB1 [21,43] (Figure 1 and Table 1).

#### 2.3.2. ABCB4 and ABCG2

In addition to ABCB1, transcriptional regulations of ABCB4 (also called MDR3) and ABCG2 are associated with miRNAs. In breast cancer cells, doxorubicin resistance is regulated by spindlin1 (SPIN1), which is an upstream regulator of ABCB4 [45]. This study provided evidence that SPIN1 is targeted by the miR-148/152 family (miR-148a-3p, miR-148b-3p, and miR-152-3p) and that the miR-148/152 family thereby effectively improves the efficacy of doxorubicin-mediated cytotoxicity [45]. Although further research is necessary to unravel the precise pathways of the SPIN1-ABCB4 axis, it might be associated with peroxisome proliferator-activated receptor alpha (PPARα) owing to the fact that PPARα is a transcription factor of ABCB4 [47] and that SPIN1-mediated activation of Akt could inactivate glycogen synthase kinase 3 beta (GSK3β), which is a negative regulator of PPARα [48,49]. Moreover, cisplatin resistance could be reversed through miR-495-3p-mediated ABCG2 suppression [46]. In this study, it was uncovered that miR-495-3p directly modulates the expression of ubiquitin-conjugating enzyme E2 C (UBE2C), a transcription factor of ABCG2 [46] (Figure 1 and Table 1).

### 2.4. A MiRNA Regulating Degradation of a Drug Transporter

A recent study has shown that miR-20a-5p is involved in the post-translational regulation of ABCB1. In miR-20a-5p over-expressing cells, the levels of mitogen-activated protein kinase 1 (MAPK1, also known as ERK2) are down-regulated, leading to the alleviation of therapeutic resistance, together with the inactivation of ribosomal protein S6 kinase (RSK) [44]. RSK is known to destabilize ubiquitin-conjugating enzyme E2 R1 (UBE2R1), thereby protecting ABCB1 from the ubiquitin–proteasomal degradation pathways [50,51] (Table 1).

## 3. MiRNAs in the Regulation of DNA Damage Repair and Therapeutic Resistance

### 3.1. DNA Damage Repair in Cancer

Effective DNA damage repair followed by cancer therapies contributes to the limited efficacy of treatments and the appearance of therapeutic resistance [2]. Since the inhibition of DNA repair pathways could make cancer cells more vulnerable to anti-cancer therapies, the elucidation of precise mechanisms of DNA damage repair could lead to the development of a strategy for more successful cancer therapy [52]. For example, temozolomide induces DNA double-strand breaks that could be repaired by the homologous recombination pathway promoted by cyclin dependent kinase 1 (CDK1) and CDK2. Indeed, pharmacological inhibition of CDK1/2 sensitizes glioblastoma cells to temozolomide [53]. A recent study demonstrated that zinc finger protein 830 (ZNF830) plays a role in the repair of DNA double-strand breaks and that knockdown of ZNF830 sensitizes lung cancer cells to olaparib, a DNA-damaging agent [54]. In addition, the importance of miRNAs in the regulation of DNA repair mechanisms has been underscored. Thus, the modulation of miRNA levels has been suggested as a therapeutic strategy to advance the efficacy of cancer treatments.

### 3.2. MiRNAs Negatively Regulating DNA Repair Mechanisms

Several miRNAs have been identified to suppress cellular factors associated with DNA repair pathways, implying a possibility that the over-expression of an miRNA acting as the repressor of DNA repair could have a therapeutic benefit in cancer.

#### 3.2.1. MiR-7-5p

Poly ADP-ribose polymerase 1 (PARP1) is known to recruit DNA repair factors to the site of DNA double-strand breaks. In lung cancer cells, the cytotoxicity of doxorubicin could be enhanced by miR-7-5p. PARP1 is down-regulated by miR-7-5p, leading to impaired DNA damage repair. Notably, miR-7-5p levels show a negative correlation with the status of doxorubicin resistance [55] (Table 2).

#### 3.2.2. MiR-30-5p

It was demonstrated that doxorubicin resistance is related to p53 mutation in breast cancer. The expression of miR-30-5p could be transcriptionally regulated by wild-type p53, and this miRNA directly targets two DNA repair elements, Fanconi anemia complementation group F protein (FANCF) and REV1 DNA directed polymerase (REV1). In p53-mutated cells, both FANCF and REV1 are abundantly expressed due to the abolishment of miR-30-5p induction, thus advancing doxorubicin resistance [56] (Table 2).

#### 3.2.3. MiR-138-5p

The excision repair cross-complementation group (ERCC) is known to participate in the nucleotide excision repair (NER) pathway [57]. Recent evidence suggested that miR-138-5p reverses drug resistance in gastric cancer cells as a result of targeting ERCC1 and ERCC4 genes [58] (Table 2). Additionally, several studies demonstrated that miR-138-5p is a tumor suppressor and suppresses the growth and metastasis of cancer cells [59,60]. However, this miRNA functions as an oncogenic miRNA in glioma stem cells via the regulation of several tumor-suppressive genes, including caspase-3 [61]. These findings indicate the complex role of miR-138-5p and suggest that further investigation into the effects of this miRNA on cellular signaling is necessary.

#### 3.2.4. MiR-182-5p and MiR-4429

BRCA1 DNA repair associated (BRCA1) functions in the repair of DNA double-stranded breaks by enhancing the recombinase activity of RAD51 recombinase (RAD51) [62]. A recent study demonstrated that miR-182-5p regulates chemosensitivity via modulation of the expression of DNA repair genes. BRCA1 and RAD51 are targeted by miR-182-5p [63,64]. In particular, the treatment of panobinostat, a histone deacetylase inhibitor, could impede DNA repair following irradiation or 2′-*C*-cyano-2′-deoxy-1-β-d-arabino-pentofuranosyl-cytosine (CNDAC) treatments by inducing miR-182-5p [63]. Furthermore, another study provided evidence that there is a negative correlation between miR-4429 levels and radioresistance in cervical cancer cells. Indeed, the up-regulation of miR-4429 improves the efficacy of irradiation by repressing RAD51, suggesting the possibility of using miR-4429 mimics to defeat radioresistance, one of the major causes of cancer recurrence [65] (Table 2).

#### 3.2.5. MiR-205-5p and MiR-211-5p

Zinc finger E-Box binding homeobox 1 (ZEB1) has been proven to be critical for the regulation of checkpoint kinase 1, which coordinates DNA damage response signaling [66]. Also, PKCε could activate the DNA-dependent protein kinase by modulating the nuclear accumulation of the epidermal growth factor receptor [67]. A recent study showed that miR-205-5p could impair DNA repair pathways by targeting ZEB1 and PKCε, leading to an elevated response to radiotherapy in prostate cancer cells [68]. Furthermore, the augmentation of carboplatin sensitivity can be achieved by miR-211-5p, owing to its ability to target multiple genes involved in DNA damage response, namely DNA polymerase eta (POLH), tyrosyl-DNA phosphodiesterase 1 (TDP1), ATRX chromatin remodeler (ATRX), mitochondrial ribosomal protein S11 (MRPS11), and ERCC excision repair 6 like 2 (ERCC6L2) [69] (Table 2).

#### 3.2.6. MiR-520g-3p and MiR-520h

The essential enzyme repairing apurinic/apyrimidinic site is apurinic/apyrimidinic endodeoxyribonuclease 1 (APEX1, also known as APE1). The depletion of APEX1 gives rise to DNA damage accumulation, eventually sensitizing cells to anti-cancer agents [70]. In multiple myeloma, levels of miR-520g-3p and miR-520h are inversely correlated with bortezomib resistance. Over-expression of these miRNAs subdues bortezomib resistance through targeting APEX1 [69] (Table 2).

### 3.3. MiRNAs Positively Regulating DNA Repair Mechanisms

By contrast, miRNAs are capable of promoting DNA damage repair and stabilizing the DNA replication fork, resulting in the aggravation of therapeutic resistance. This indicates that the knockdown of a resistance-associated miRNA could have a therapeutic benefit in cancer.

#### 3.3.1. MiR-488-3p

Eukaryotic translation initiation factor 3 subunit A (EIF3A) has been demonstrated to down-regulate NER activity by regulating the levels of NER factors. Therefore, the knockdown of EIF3A interrupts DNA damage-induced cell death [71,72]. A recent study revealed that miR-488-3p levels are higher in cisplatin-resistant lung cancer cells than in parental cells, and NER is activated by miR-488-3p, which targets EIF3A [73] (Table 2).

#### 3.3.2. MiR-493-5p

Stabilization of the DNA replication fork is one of the causes of therapeutic resistance. For example, the depletion of a chromatin-remodeling factor, such as chromodomain helicase DNA binding protein 4 (CHD4), confers cisplatin resistance in BRCA2-mutated cancer cells [74]. A similar conclusion was reached by investigating the role of miR-493-5p. Resistance to cisplatin and olaparib is mediated by miR-493-5p over-expression in BRCA2-mutated ovarian cancer cells because of the ability of miR-493-5p to target multiple genes (e.g., CHD4) involved in regulating single-strand annealing DNA repair and genomic stability [75] (Table 2).

## 4. Autophagy-Regulating MiRNAs and Therapeutic Resistance

### 4.1. General Mechanisms of Autophagy

Autophagy is a highly conserved cellular process by which cytoplasmic materials are isolated into double-membrane autophagosomes that fuse with lysosomes. The autophagic degradation activity (also known as autophagic flux) is known to maintain cellular homeostasis and protein/organelle quality control [76]. Several components are involved in the machinery of autophagy. Generally, mTOR complex 1 (mTORC1) represses unc-51-like autophagy-activating kinase (ULK), thus leading to the inhibition of autophagy induction. After being released from the inhibitory actions of mTORC1, the ULK complex, including the FAK family-interacting protein of 200 kDa (FIP200) and autophagy-related 13 (ATG13), is activated. Ultimately, the activated ULK complex induces autophagy by phosphorylating BECLIN1 and its binding partners, VPS34 (also known as phosphatidylinositol 3-kinase catalytic subunit type 3, PIK3C3) and ATG14 [77,78]. ATG12 is conjugated to ATG5 via ATG7, and then the ATG12-ATG5 conjugate forms a complex with ATG16L1. The ATG5–ATG12–ATG16L1 complex facilitates the lipidation of LC3I to LC3II, which is required for autophagosome formation [79]. Since multiple factors are necessary to accomplish autophagic processes, autophagy can be inhibited at several points.

### 4.2. Dual Roles of Autophagy in Cancer

In the context of cancer, autophagy plays a pivotal role in the regulation of cellular events, such as cell death, cancer stemness, and therapeutic resistance. Regarding cancer therapy, strategies of both stimulation and inhibition of autophagy have been considered. For example, the silencing of ATG5 and ATG12 could inhibit autophagy, consequently attenuating CSC properties [6,80]. The blockage of autophagic flux leads to cell death by augmenting stress-activated proteins, such as JNK and p38 [81]. It was also reported that treatment with paclitaxel can lead to the activation of autophagy, which, in turn, confers resistance to paclitaxel. Therefore, the inhibition of BECLIN1 enhances the anti-cancer activity of paclitaxel via the attenuation of cytoprotective autophagy in ovarian cancer cells [3]. Likewise, cell death induced by an epigenetic agent could be stimulated by suppressing BECLIN1 expression in drug-resistant leukemia stem cells [82].

By contrast, autophagy triggered by caloric restriction mimetics was proposed to deplete regulatory T cells, reinforce cancer immunosurveillance, and improve the efficacy of mitoxantrone as well as oxaliplatin [83]. Such discrepancy could be due to the fact that cellular factors involved in the process of autophagy regulate other cellular responses, such as cytokinesis, endocytosis, cell growth, and cell death, in an autophagy-independent manner, and that the effects of autophagy on the fate of cancer cells are dependent on p53 status [84,85,86]. In this review, we focus on the inhibitory roles of miRNAs in cytoprotective autophagy along with their effects on chemotherapeutic agents.

### 4.3. MiRNAs Regulating mTOR and mTORC1

As mentioned above, autophagy is negatively regulated by mTORC1. An association between mTORC1-dependent autophagy and miRNAs has been investigated in conjunction with the anti-cancer activity of therapeutic agents. Nuclear factor erythroid 2 like-2 (NRF2) is a mediator of therapeutic resistance to Trichostatin A. A mechanism underlying NRF2-mediated resistance suggested that NRF2 could stimulate autophagy induction following Trichostatin A treatment via transcriptionally up-regulating the levels of miR-129-3p, which targets mTOR [87] (Table 3). Actually, two functional miRNAs (miR-5p and miR-3p) can be derived from the same miRNA precursor [88]. Since both miR-129-3p and miR-129-5p (see Table 1) are derived from the same precursor, and both mature miRNA strands are functionally different in cells, it is feasible that over-expression of the miR-129 precursor in cancer cells using vector systems triggers cytoprotective autophagy, but inhibits ABCB1 levels.

Another recent study showed that RAB12 member RAS oncogene family (RAB12) is an endogenous mTORC1 inhibitory factor and protects cancer cells from drug-induced cell death by activating cytoprotective autophagy. RAB12 is targeted by miR-148-3p; therefore, the introduction of miR-148-3p in cancer cells can reverse therapeutic resistance [89] (Table 3). Since miR-148-3p negatively controls both cytoprotective autophagy and ABCB4 expression (see Table 1), this implies the possibility of the potential therapeutic application of miR-148-3p to suppress CSC properties.

### 4.4. MiRNAs Regulating ULK1, BECLIN1, and ATG14

Chemosensitization could be induced by miRNAs via regulating the levels of ULK1, BECLIN1, and ATG14. In hepatocellular carcinoma cells, miR-26-5p targets ULK1, leading to the enhanced cytotoxicity of doxorubicin [91]. Besides, BECLIN1 and ATG14 are post-transcriptionally regulated by miR-409-3p and miR-152-3p, respectively. Consequently, these miRNAs can potentiate the cytotoxicity of therapeutic agents [98,101] (Table 3). As mentioned above, miR-152-3p also regulates the levels ABCB4 (Table 1), implying a potential role of this miRNA in regulating therapeutic resistance.

### 4.5. MiRNAs Regulating HMGBs

The possibility of miRNA-mediated chemosensitization was also associated with the function of high mobility group box 1 (HMGB1) and HMGB2. In terms of autophagy, HMGB1 is known to promote autophagy by preventing the interaction between Bcl-2 and BECLIN1 in the cytoplasm [105]. A reduced form of HMGB1 in the extracellular space also facilitates BECLIN1-dependent autophagy [106]. Direct inhibition of HMGB1 by miR-34-5p and miR-410-3p impedes autophagy induction, reversing resistance to several chemotherapeutic agents [92,102]. Likewise, HMGB2 is presumed to promote autophagy by the same mechanisms as HMGB1. Negative regulation of HMGB2 levels in miR-23-3p over-expressing cells is one of the causes of autophagy inhibition [90] (Table 3).

### 4.6. MiRNAs Regulating the Lipidation of LC3

In the case of miR-23-3p, another identified target is ATG12. This further provides a mechanism whereby miR-23-3p sensitizes cancer cells to chemotherapeutic agents [90]. It was also found that ATG12 is directly targeted by miR-214-3p [99]. This association between ATG12 and miR-214-3p explains the role of this miRNA in promoting radio-sensitivity, since the exposure of irradiation can induce ATG12-mediated autophagy, which, in turn, leads to radio-resistance [99] (Table 3). However, miR-214-3p protects lung cancer cells from radiotherapy-induced cell death [107], indicating that the overall effects of miR-214-3p on cancer therapeutics could be distinct in a cellular context-dependent manner.

Additional studies have also shown the effects of autophagy-regulating miRNAs on therapeutic agents. ATG7, a mediator of ATG12-ATG5 conjugation, is targeted by miR-520-3p, and over-expression of miR-520-3p sensitizes drug-resistant cells to chemotherapy [103]. ATG5 can be targeted by multiple miRNAs, including miR-137-3p, miR-142-3p, and miR-224-3p. The ability of these miRNAs to restrain autophagy contributes to the sensitization of cancer cells to anti-cancer agents [95,97,100]. In addition to ATG5, the inhibitory effects of miR-142-3p on therapeutic resistance can be mediated by targeting ATG16L1 [97]. Moreover, a recent study also indicated that autophagy inhibition in miR-874-3p over-expressing cells abrogates therapeutic resistance [104] (Table 3).

### 4.7. MiRNAs and Other Autophagy-Regulating Genes

The detection of dysregulated miRNAs in cancer tissues identified that miR-140-5p is down-regulated in the chemo-resistant osteosarcoma tissues, and targets high-mobility group nucleosome binding domain 5 (HMGN5), which is a positive regulator of BECLIN1 and ATG5 [96]. Moreover, it was described that miR-101-3p blocks autophagy induction via the down-regulation of ATG4D, RAB5A, and Stathmin 1 (STMN1) [93,94] (Table 3). Autophagy, promoted by ATG4D, protects cells from stress-induced apoptosis [108]. RAB5A is associated with autophagosome formation and ERK-mediated autophagy [109,110]. The mechanisms by which STMN1 controls autophagy are not fully understood, but STMN1 positively regulates basal and rapamycin-induced autophagy in cancer cells [94].

## 5. MiRNAs Involved in the Regulation of Therapeutic Resistance Associated with Cancer Stemness

### 5.1. Cancer Stem Cells

Cancer stem cells (CSCs), also known as cancer stem-like cells and tumor-initiating cells, have been identified in cancers, and they have notable characteristics of stemness, such as self-renewal and multilineage differentiation capabilities. Several signaling pathways, such as Wingless (Wnt)/β-catenin, Notch, Janus kinase (JAK)/signal transducer and activator of transcription 3 (STAT3), Hedgehog, nuclear factor kappa B (NF-κB), ERK, and PI3K/Akt, regulate the induction and maintenance of CSC properties [111,112]. In terms of cancer therapies, CSCs are known to be resistant to anti-cancer treatments, as discussed in the introduction. Therefore, targeted therapy against signaling pathways involved in CSCs has been suggested to improve the anti-cancer activity of therapeutic agents [4,113]. Indeed, accumulating evidence from recent studies evidently demonstrates the noticeable effects of miRNAs on CSC-associated signaling pathways (Figure 2 and Table 4).

### 5.2. MiRNAs Regulating Wnt/β-Catenin Signaling

#### 5.2.1. Wnt/β-Catenin Signaling and CSCs

Wnt/β-catenin signaling is associated with CSC properties. Activation of Wnt/β-catenin signaling orchestrates the self-renewal of CSCs and induces several stemness factors, such as OCT4, SOX2, and CD44 [114,115]. Members of the secreted frizzled-related protein (SFRP) family are Wnt antagonists and inactivate Wnt ligands, impairing signaling pathways from FZD receptors and low-density lipoprotein receptor-related proteins (LRPs) [114].

#### 5.2.2. Wnt/β-Catenin Signaling-Regulating MiRNAs

A recent study indicated that miR-93-3p and miR-105-5p are up-regulated in triple-negative breast cancer (TNBC) tissues, compared to non-TNBC tissues. Both miR-93-3p and miR-105-5p directly target SFRP1, thereby inducing the enhanced stemness of TNBC cells and resistance to cancer therapies via activating Wnt/β-catenin signaling [116]. Also, the cancer stemness of oral squamous cell carcinoma cells is enhanced by miR-1246, which targets cyclin G2 (CCNG2) [117]. CCNG2 serves as a negative regulator of Wnt/β-catenin signaling by modulating the expression of dishevelled segment polarity protein 2 (DVL2), LRP6, and β-catenin [118]. Indeed, knockdown of miR-1246 leads to a reduction in SOX2 expression [117]. In addition, a tumor suppressive miR-124-3p was identified to inhibit the self-renewal of non-small-cell lung cancer cells as a consequence of targeting ubiquitin specific peptidase 14 (USP14), which is a positive regulator of β-catenin expression [119,120] (Table 4).

### 5.3. MiRNAs Regulating Notch Signaling

#### 5.3.1. Notch Signaling and CSCs

Since Notch signaling functions as one of the potential mediators of CSC maintenance, the targeted inhibition of Notch signaling has been considered as a strategy to eradicate CSC populations. For example, an application of pharmacological or endogenous inhibitors of Notch receptors down-regulates stemness factors (e.g., OCT4) and potentiates the efficacy of anti-cancer agents, including cisplatin and sorafenib, in renal cell carcinoma cells [121].

#### 5.3.2. MiRNAs Regulating Notch Receptor 1 and 3

Notch signaling is also affected by miRNAs that post-transcriptionally regulate the levels of Notch receptors. Notch 1 is targeted by miR-34-5p and miR-139-5p in breast cancer and colorectal carcinoma cells, respectively [122,123]. In doxorubicin-resistant breast cancer cells, miR-34-5p is down-regulated compared to parental cells, implying the possibility that this miRNA is pertinent to therapeutic resistance. In addition, over-expression of miR-34-5p represses the self-renewal capacity of breast cancer stem cells [122]. In colorectal cancer cells, expression of miR-139-5p is lower in oxaliplatin- and vincristine-resistant carcinoma cells than in non-resistant cells. These drug-resistant cells have more CD44- and CD133-positive populations than non-resistant cells. As expected, the over-expression of miR-139-5p renders CD44- and CD133-positive cells sensitive to several anti-cancer drugs [123]. Furthermore, Notch 3 is directly targeted by miR-136-5p; therefore, CSC spheroid formation is restricted by this miRNA [124] (Table 4).

#### 5.3.3. A MiRNA-Regulating Notch Receptor 2 and its Downstream Signaling Factor

Recombination signal-binding protein for immunoglobulin kappa J region (RBPJ) is the transcription mediator of Notch signaling. Upon ligand binding, the Notch intracellular domain (NICD) is released from the plasma membrane and interacts with RBPJ to activate target genes [125]. It was demonstrated that miR-195-5p suppresses the levels of stemness factors (e.g., CD133 and SOX2) in colorectal CSCs by targeting Notch 2 and RBPJ [126] (Table 4).

### 5.4. MiRNAs Regulating JAK/STAT3 Signaling

#### 5.4.1. JAK/STAT3 Signaling and CSCs

JAK/STAT3 signaling contributes to CSC maintenance, as well as therapeutic resistance. For instance, the inhibition of JAK/STAT3 signaling with ruxolitinib results in a reduction in CSC hallmarks, such as spheroid formation capacity [127]. JAK/STAT3 signaling also induces carnitine palmitoyltransferase 1B (CPT1B) and activates the fatty acid beta oxidation (FAO) pathway, which is a critical regulator of CSC self-renewal. Indeed, an inhibition of JAK/STAT3 and FAO pathways sensitizes cells to chemotherapies [128].

#### 5.4.2. MiRNAs Regulating Negative Regulators of JAK/STAT3 Signaling

JAK/STAT3 pathways could be modulated by miRNAs, eventually affecting the therapeutic efficacy of anti-cancer agents. For instance, miR-196-5p is highly expressed in colorectal cancer tissues compared to non-cancerous tissues. Both the stemness and therapeutic resistance of colorectal cancer cells are promoted by miR-196-5p. It was identified that miR-196-5p directly targets the suppressor of cytokine signaling 1 and 3 (SOCS1 and SOCS3), which are negative regulators of JAK/STAT3 signaling [129].

The transcriptional activity of STAT3 is hampered by protein tyrosine phosphatase non-receptor types (PTPNs) [130]. A recent study demonstrated that multiple negative regulators of JAK/STAT3 signaling, including SOCS2, SOCS5, PTPN1, and PTPN11, are known to be directly suppressed by miR-589-5p. Therefore, CSC properties can be reinforced by this miRNA in hepatocellular carcinoma cells [131] (Table 4).

### 5.5. A MiRNA and Hedgehog Signaling

Activation of Hedgehog (Hh) signaling is triggered by the binding of Hh ligands to patched (PTCH) receptors. Interactions of Hh ligands with PTCH receptors stimulate smoothened (SMO) and glioma-associated oncogene (GLI) transcription factors [132,133]. The inhibition of SMO and GLI decreases the number of aldehyde dehydrogenase (ALDH)-positive CSCs in melanoma [134]. This indicates the significance of Hh signaling in CSC maintenance. A recent finding indicated that miR-324-5p could directly control SMO and GLI1 expression, and subsequently inhibit the colony formation of multiple myeloma cells in a stem cell medium [135] (Table 4).

### 5.6. MiRNAs Regulating NF-κB Signaling and PD-L1

NF-κB signaling regulates multiple cellular processes, including metastasis and CSC properties [136]. A recent study demonstrated that miR-423-5p targets inhibitor of growth 4 (ING4), a negative regulator of NF-κB. Therefore, this miRNA contributes to the augmented expression of glioma stem cell factors, including CD133 and SOX2 [137] (Table 4). Recently, miR-423-5p was also reported to promote the metastasis of lung adenocarcinoma and gastric cancer cells [138,139], implying that knockdown of miR-423-5p may have a therapeutic benefit in several cancer types.

Another interesting finding regsrding CSC maintenance is that programmed death ligand 1 (PD-L1, also known as CD274) could stimulate CSC expansion via the high mobility group A (HMGA)-dependent Akt and ERK pathways [140]. PD-L1 in CSCs also confers immune evasion, thereby sustaining the tumorigenesis of CSCs [141]. Recently, miR-873-5p was shown to target PD-L1, thus attenuating breast cancer stemness [142] (Table 4).

### 5.7. Other MiRNAs Directly or Indirectly Regulating Stemness Factors

#### 5.7.1. Direct Regulation of Stemness Factors

Stemness factors are bona fide targets of miRNAs. In colorectal cancer cells, miR-450b-5p represses cancer stemness via targeting SOX2 [143]. Furthermore, tumor-suppressive miR-145-5p targets CD44, thereby suppressing stemness and therapeutic resistance in gastric cancer cells [144]. Of note, miR-145-5p also targets other stemness factors, including KLF4, OCT4, and c-MYC, hence sensitizing colorectal cancer cells to anti-cancer therapies [145] (Figure 2 and Table 4). However, there is an ongoing debate on the tumor-suppressive functions of miR-145-5p since it was discovered that miR-145-5p could act as an oncogenic miRNA in genetically engineered mouse models of cancer by promoting angiogenesis [146], implying the possibility of different impacts of miR-145-5p on the fate of cells between species. Such a discrepancy demonstrates the need for caution in the selection of miR-145-5p as a candidate miRNA for miRNA-based combinatorial cancer therapy. Interestingly, the pool of predicted target genes is dissimilar between humans and mice based on a target prediction database, although sequences of mature miR-145-5p are identical. Further studies are required to unveil the precise role of this miRNA.

#### 5.7.2. Indirect Regulation of Stemness Factors

Cancer stemness is also maintained by other factors, such as Golgi phosphoprotein 3 (GOLPH3), yin and yang 1 (YY1), and neurofilament light (NEFL). These factors are capable of modulating the expression levels of various stemness factors in a positive manner.

##### GOLPH3

In bladder cancer cells, the over-expression or knockdown of GOLPH3 increases or decreases the expression of CSC factors (CD44, KLF4, ALDH1, and SOX2), respectively. The levels of miR-34-5p are down-regulated in gemcitabine- and cisplatin-resistant bladder cancer cells, contributing to enriched CSC populations [147] (Figure 2 and Table 4). Furthermore, the ability of miR-34-5p to target multiple genes (see Table 3 and Table 4) can explain why this miRNA functions effectively with several chemotherapeutics.

##### YY1

Further evidence indicates that YY1 regulates CSC hallmarks of glioblastoma. The depletion of YY1 preferentially down-regulates levels of CD133 and NESTIN. Also, miR-7-5p levels, which target YY1, are remarkably decreased in temozolomide-resistant glioblastoma cells. Therefore, miR-7-5p is considered as a stemness- and chemoresistance-suppressive miRNA [148]. Additionally, miR-186-5p, which is down-regulated in cisplatin-resistant glioblastoma cells, targets YY1, weakens the sphere formation of glioblastoma cells, and improves the efficacy of chemotherapeutic agents [149] (Figure 2 and Table 4).

##### NEFL

Several stemness factors are regulated by the neurofilament light polypeptide (NEFL). In fact, miR-381-3p contributes to temozolomide resistance in glioblastoma cells by targeting NEFL and regulating stemness factors in an NEFL-dependent manner [150] (Figure 2 and Table 4).

## 6. MiRNAs Involved in the Regulation of Therapeutic Resistance Associated with Epithelial-Mesenchymal Transition (EMT)

### 6.1. EMT and Cancer Stemness

EMT is known to confer CSC properties on cancer cells. For example, transforming growth factor-beta (TGF-β) and hepatocyte growth factor (HGF) signaling often induces EMT by activating EMT transcription factors and enhances the stemness of cancer cells [7,151,152]. This indicates that inhibition of the mechanisms underlying EMT can improve therapeutic responses by blocking the transformation of cancer cells to the CSC state.

### 6.2. MiRNAs Regulating TGF-β Signaling

Owing to the effort to screen EMT-regulating miRNAs, miR-509-5p was identified to repress the EMT process by targeting vimentin (VIM) and HMGA2, both of which are positive regulators of TGF-β signaling [153]. In this study, miR-1243 was also demonstrated to inhibit the EMT process by targeting SMAD family member 2 (SMAD2) and SMAD4. Certainly, over-expression of either miR-509-5p or miR-1243 augments gemcitabine efficacy [153]. Recently, it was demonstrated that treatment of 5-fluorouracil reduces the expression of miR-204-5p, which was identified to target TGF-β receptor 2 (TGFBR2) [154] (Table 5).

Integrins are known to induce the transmembrane signaling pathways to modulate the EMT process. For instance, it was noted that the translational activation of integrin subunit β 3 (ITGB3) supports TGF-β pathways and advances malignant phenotypes, such as EMT. Thus, targeting integrins with antibodies has been attempted for cancer therapy [155,156]. A recent study of miR-483-3p indicated that this miRNA enhances the effectiveness of gefitinib by targeting ITGB3 [154] (Table 5). Although multiple signaling pathways coordinately regulate cancer stemness, the dysregulation of miR-483-3p may play a partial role in the ITGB3-mediated regulation of stemness in cancer, since ITGB3 was reported to drive cancer stemness [157].

### 6.3. MiRNAs and HGF/c-MET

HGF induces EMT, leading to the development of therapeutic resistance to gefitinib in lung cancer cells [158]. In this study, it was found that both miR-1-3p and miR-206 target c-MET, a receptor for HGF, thus impairing HGF-induced EMT and improving the efficacy of anti-cancer therapy. In addition, c-MET receptors orchestrate the EMT process by regulating miRNA levels. For example, the inhibition of c-MET receptors increases the levels of miR-103-3p and miR-203a-3p, both of which hamper the EMT process and intensify the efficacy of gefitinib in lung cancer cells [159] (Table 5). Interestingly, miR-103-3p exerts an oncogenic role in different types of cancer. For example, miR-103-3p is known to enhance the cell proliferation of gastric cancer cells [160], indicating that miR-103-3p-based cancer therapy requires careful consideration of cancer type.

### 6.4. MiRNAs Directly Regulating EMT-Related Transcription Factors and Markers

The chemosensitization effects of miR-204-5p are also mediated by targeting ZEB1 in addition to TGFBR2 [165]. Besides, it has been shown that the expression of EMT-promoting transcription factors is directly modulated by other miRNAs. In ovarian cancer cells, miR-363-3p targets snail family transcriptional repressor 1 (SNAI1). As a consequence, the over-expression of miR-363-3p in cancer cells reverses EMT-mediated drug resistance [166]. In addition, the regulation of ZEB1 expression is mediated by miR-574-3p and miR-708-3p in gastric and breast cancer cells, respectively. Both miRNAs negatively regulate the process of EMT and render cancer cells susceptible to chemotherapy agents [133,167]. In the case of miR-708-3p, it also regulates EMT markers, cadherin 2 (CDH2, also known as N-cadherin) and VIM, by directly binding to their 3′ UTRs [133]. In addition, miR-128-3p and miR-873-5p were identified to make cancer cells more responsive to anti-cancer agents by suppressing ZEB1 expression [162,169] (Table 5). ZEB1 can transcriptionally induce expression of PD-L1 [170]. Therefore, miR-873-5p has a great possibility of potentially regulating ZEB-1/PD-L1 axis, since this miRNA also targets PD-L1 (see Table 4).

The miR-200 family is composed of miR-200a, miR-200b, miR-200c, miR-141, and miR-429. It was demonstrated that miR-200c is down-regulated in TGF-β-treated or trastuzumab-resistant gastric cancer cells and targets ZEB1 and ZEB2. Therefore, over-expression of miR-200c can reverse trastuzumab resistance in gastric cancer cells [164]. Moreover, the over-expression of miR-200 family (miR-200a, miR-200b, and miR-429) was found to block the EMT process, thereby reversing therapeutic resistance, rather than impacting lung metastasis in an orthotopic breast tumor model [163] (Table 5).

### 6.5. MiRNAs Indirectly Regulating EMT-Related Transcription Factors

Death-effector domain-containing DNA-binding protein (DEDD) inversely controls the process of EMT by attenuating the expression of EMT-promoting elements, such as SNAI1 and Twist family BHLH transcription factors (Twist) [171]. In gastric cancer cells, a reduction in miR-17-5p interrupts the EMT process by up-regulating its target, DEDD, thereby affecting the therapeutic resistance of gastric cancer cells [161] (Table 5).

## 7. Extracellular Vesicle MiRNAs and Therapeutic Resistance

### 7.1. Extracellular Vesicles (EVs)

There is mounting evidence that EVs (exosomes and microvesicles) play a critical role in the intercellular communication by transferring cargo molecules, such as cytosolic proteins, lipids, and RNA [172]. Exosomes are released into extracellular space by fusion of multivesicular endosomes with the cell membrane, whereas microvesicles are developed directly from the cell membrane [172]. It has been demonstrated that EVs from neighboring cells can affect various biological properties of cancer cells, such as proliferation, metastasis, hypoxia tolerance, and therapeutic resistance [173,174]. For example, mitochondrial DNA in EVs derived from cancer-associated fibroblasts (CAFs) contributes to the development of resistance to hormone therapy in breast cancer [175]. Also, cancer cells exposed to anti-cancer agents can deliver EVs harboring drug efflux and pro-survival proteins into circumjacent cancer cells, thereby playing a part in enhancing cell survival and therapeutic resistance [176,177].

### 7.2. EVs from CAAs, TAMs, and CAFs

In addition to DNA and proteins in EVs, miRNAs are also enriched in EVs derived from cancer-associated cells, hence affecting the expression of cellular factors involved in therapeutic resistance.

#### 7.2.1. MiR-21-5p

Exosomes derived from cancer-associated adipocytes (CAAs), tumor-associated macrophages (TAMs), and CAFs within the cancer microenvironment contain miR-21-5p, and exosome-mediated miR-21-5p delivery into cancer cells confers therapeutic resistance in multiple types of cancer. It could be due to the ability of miR-21-5p to target tumor-suppressive genes, such as phosphatase and tensin homolog (PTEN) and apoptotic peptidase-activating factor 1 (APAF1) [8,178] (Figure 3).

#### 7.2.2. MiR-196a-5p

In head and neck cancers, exosomal miR-196a-5p, derived from CAFs, contributes to cell proliferation and cisplatin resistance. By targeting cyclin-dependent kinase inhibitor 1B (CDKN1B) and inhibitor of growth family member 5 (ING5), miR-196a-5p facilitates cell growth and inhibits cell death, respectively. Therefore, exosomal miR-196a-5p ultimately serves as the deteriorating factor of cisplatin resistance [179] (Figure 3).

### 7.3. EVs from Drug-Resistant Cancer Cells and CSCs

It has been underscored that drug-resistant cancer cells also secrete EVs to transfer miRNAs into adjacent cancer cells and that transferred miRNAs contribute to the development of therapeutic resistance in non-resistant cells.

#### 7.3.1. MiR-32-5p

The enriched expression of miR-32-5p, which targets PTEN, was identified in exosomes secreted from 5-fluorouracil-resistant hepatocellular carcinoma cells. The treatment of non-resistant cancer cells with these exosomes confers therapeutic resistance to several agents, such as 5-fluorouracil, oxaliplatin, gemcitabine, and sorafenib, owing to the downregulation of PTEN and the activation of PI3K/Akt pathways [180] (Figure 3).

#### 7.3.2. MiR-155-5p

Exosomes harboring miR-155-5p are secreted by paclitaxel- or doxorubicin-resistant gastric and breast cancer cells, as well as breast CSCs. Delivery of these exosomes can convert the phenotype of drug-sensitive cells to drug-resistant cells through miR-155-5p-mediated alteration of the levels of negative regulators of EMT, including GATA binding protein 3 (GATA3), tumor protein p53-inducible nuclear protein 1 (TP53INP1), CCAAT enhancer-binding protein beta (CEBPB), and forkhead box O3A (FOXO3A) [181,182] (Figure 3). However, miR-155-5p acts as a tumor-suppressive miRNA and sensitizes multiple myeloma cells to bortezomib [9], indicating that either the replacement or knockdown of this miRNA can be applied for cancer therapy with other anti-cancer agents, depending on the type of cancer.

#### 7.3.3. MiR-222-3p and MiR-365

Similarly, exosomes derived from gemcitabine-resistant lung cancer and imatinib-resistant chronic myeloid leukemia (CML) cells contain miR-222-3p and miR-365, respectively [183,184]. Exosomes containing miR-222-3p are transferred to other cancer cells, where miR-222-3p directly regulates SOCS3 and develops malignant phenotypes of lung cancer cells along with gemcitabine resistance [183]. In addition, the therapeutic resistance to imatinib could be triggered by exosomal miR-365, which targets Bcl-2 associated X (BAX) in CML cells [184] (Figure 3).

#### 7.3.4. MiR-432a-5p, MiR-486-3p, and MiR-501-5p

Consistent with the above observations, other studies found that exosomal miRNAs, miR-432a-5p, miR-486-3p, and miR-501-5p, could be derived from drug-resistant cancer cells, and these miRNAs develop the resistance in surrounding cells [185,186,187]. Exosomal miR-432a-5p transmits palbociclib resistance to other cancer cells via down-regulating SMAD4 and subsequently inducing CDK6 expression in breast cancer cells [185]. In lung adenocarcinoma, one of the differentially expressed miRNAs in EVs released from drug-resistant cells is miR-486-3p. The delivery of miR-486-3p into surrounding cancer cells causes a down-regulation of PTEN and a development of resistance toward tyrosine kinase inhibitors [186]. Also, by targeting the BH3-like motif containing an inducer of cell death (BLID), exosomal miR-501-5p activates Akt signaling and represses caspase activation in recipient cells, eventually contributing to doxorubicin resistance in gastric cancer cells [187] (Figure 3). Overall, these results clearly indicate that knockdown of extracellular vesicle miRNAs has a therapeutic benefit in several cancer types.

## 8. Conclusions

Resistance to anti-cancer treatments is intricate, and divergent mechanisms are involved and act together. In light of this, the simultaneous inhibition of signaling components has been considered to improve the current condition of cancer treatment. For instance, a recent study demonstrated that combined targeting of the Hh pathway and autophagy efficiently inhibits cell proliferation and induces cell death in drug-resistant cells [188], providing evidence that combination therapy is a reasonable approach to enhance therapeutic efficacy in cancers. However, the occurrence of therapeutic resistance continues to be a significant cause of the failure of current combination therapy. It has been demonstrated that the acquisition of resistance can occur following combination treatment and that the use of a third agent can address the emergence of resistance by inhibiting a factor involved in the resistance [10,11,189]. This implies that the multiple targeting of resistance-mediating factors is better than the current approach to increase the cellular response to cancer therapy and that the development of new combination strategies is still required to improve treatment outcomes.

A growing number of investigations have successfully demonstrated that single miRNA can control the effectiveness of anti-cancer treatments by regulating the expression of resistance-related factors, as highlighted in this article. Most of the investigations have identified a single target gene for an miRNA, implying that the identified target of an miRNA at least partly accounts for the regulation of efficacy of cancer therapeutics. Since an miRNA can regulate diverse biological pathways owing to its multiple targets, it is probable that there are other identified/unidentified targets, which modulate multiple resistance-related signaling pathways. Further research is required to address the extensive miRNA–target gene interactions. Additionally, there are some considerations relating to the characteristic features of miRNAs when designing miRNAs for therapeutic purposes.

Some miRNAs have dual roles, depending on cancer type, since they can target both oncogenes and tumor suppressors. Therefore, the appropriate selection of miRNAs that are particularly relevant to the specific cancer type is necessary to achieve a better response to treatments through miRNA-based combinatorial cancer therapy. For this purpose, further investigations are required to comprehensively identify the dysregulated miRNAs in therapy-resistant cancer cells and tissues. In addition, miRNAs that influence the efficacy of anti-cancer therapies could also influence the physiological processes of non-cancerous cells. For example, miR-138-5p was proposed to negatively control the osteogenic differentiation of human mesenchymal stem cells [190]. In the case of miR-155-5p, knockdown of this miRNA induces the levels of inflammatory cytokines [191]. This emphasizes the need to consider cancer-specific modulation of miRNA levels to avoid unintended side effects of miRNA-based combinatorial cancer therapy. Indeed, the targeted delivery of miRNA modulators to cancer cells has been developed to minimize feasible side effects [192,193].

A recent study evidently demonstrated that vector-based miRNA expression could generate both miR-202-3p and miR-202-5p [194]. In this study, miR-202-3p inhibits the proliferation of colorectal cancer cells, but miR-202-5p has no effect on cell proliferation. Although miR-202-5p is not associated with the growth regulation of colorectal cancer cells, another study demonstrated that miR-202-5p regulates the sensitivity of doxorubicin in breast cancer cells [195]. As discussed in Section 4.3, miR-129-3p and miR-129-5p also have opposite functions in the regulation of drug efficacy. Therefore, functional differences between miR-3p and miR-5p are an additional concern that should be considered when evaluating the effects of miRNAs on other anti-cancer therapies. In addition, the modulation of single miRNA expression could only partially regulate signaling pathways that regulate the cellular response to cancer therapeutics since (1) over 2000 mature miRNAs have been described in miRBase [196], (2) multiple miRNAs function cooperatively with other miRNAs [197], (3) the same gene is undoubtedly regulated by multiple miRNAs, and (4) alternative pathways can be induced by the modulation of miRNA expression owing to the direct and/or indirect effects of a miRNA on transcription factors [198]. Therefore, it appears essential to investigate whether simultaneous replacement or knockdown of functionally relevant miRNAs is more potent than single miRNAs for miRNA-based combinatorial cancer therapy.

A considerable number of studies have been conducted and demonstrated the benefit of miRNA-based combinatorial cancer therapy in combating cancer. However, it is essential to carefully consider several characteristic features of miRNAs, including their dual roles, side effects, and the functional differences between miR-3p and miR-5p, when designing and evaluating the efficacy of miRNA-based combinatorial cancer therapy. Further investigation of miRNA’s target genes is also necessary to comprehensively elucidate the functions of miRNAs and to analyze miRNA-signaling pathway networks in cells. Advanced knowledge of the multifactorial nature of miRNAs will enable miRNA-based combinatorial cancer therapy to move toward clinical application in the future.

## Figures and Tables

**Figure 1 cells-09-00029-f001:**
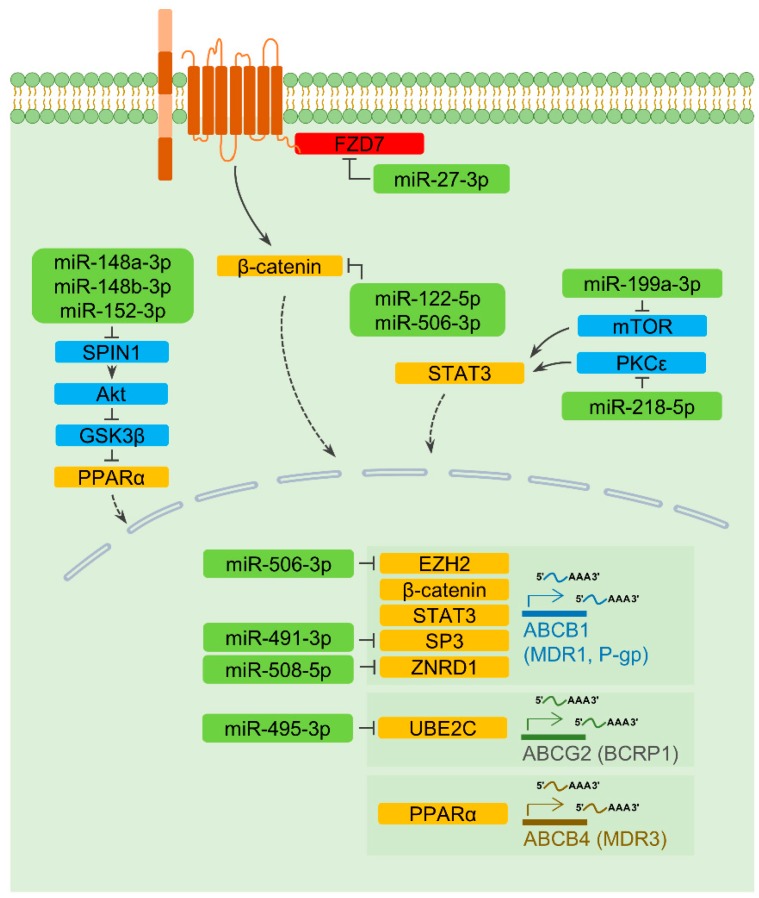
Micro-RNA (MiRNA)-mediated regulation of the expression of drug transporters. Rounded rectangles indicate miRNAs (light green), transcription factors (orange), cytoplasmic signaling molecules (light blue), and a transmembrane receptor (red). Activation is denoted by solid line arrows, and inhibitory effects are indicated by perpendicular lines. Dashed arrows represent the nuclear translocation of transcription factors. Several miRNAs impact the efficacy of cancer therapeutic agents by transcriptionally regulating the levels of drug transporters, as described in Section 2.3.

**Figure 2 cells-09-00029-f002:**
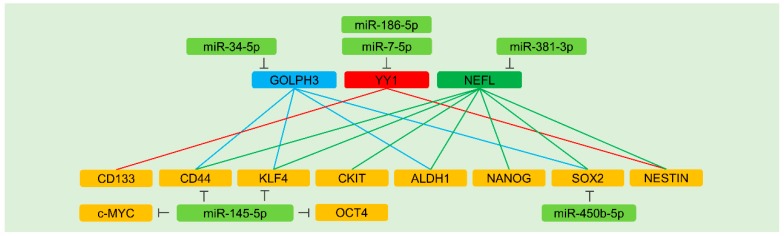
MiRNA-mediated regulation of the factors associated with cancer stemness. Rounded rectangles denote miRNAs (light green), stemness factors (orange), and upstream regulators of stemness factors (light blue, red, and green). Inhibitory effects are indicated by perpendicular lines. The positive regulation of stemness factors by each upstream factor is represented by solid lines. The effects of miRNAs on anti-cancer therapies are described in Section 5.7.2 and Table 4.

**Figure 3 cells-09-00029-f003:**
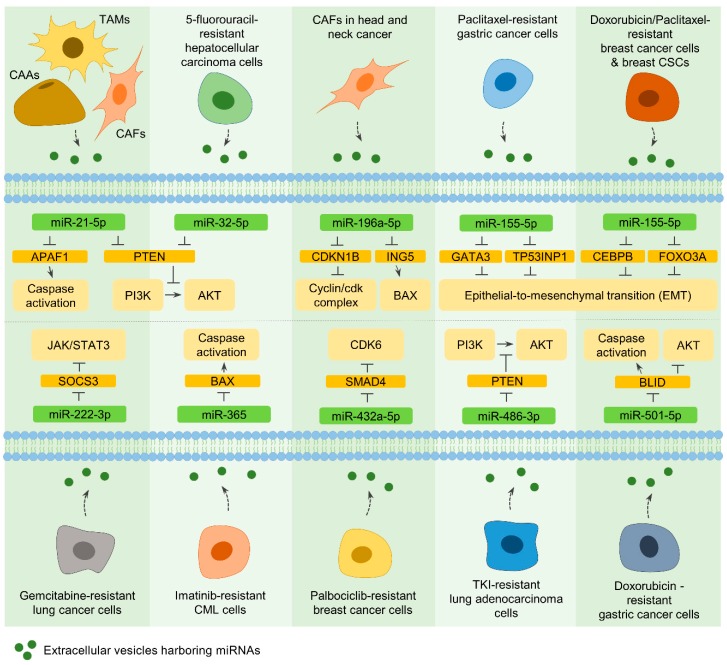
Extracellular vesicle miRNAs are responsible for therapeutic resistance. Exosomes and microvesicles (EVs), derived from cancer-associated cells, drug-resistant cancer cells, and cancer stem cells (CSCs), confer neighboring cells resistance to various anti-cancer treatments via transferring miRNAs, which regulate several factors associated with resistance mechanisms. Rounded rectangles represent miRNAs (light green), miRNA targets (orange), and signaling factors/cellular events affected by miRNA targets (light orange). Activation is indicated by solid line arrows, and inhibitory effects are demonstrated by perpendicular lines. The secretion of extracellular vesicles is denoted by dashed arrows. Potential mechanisms underlying the role of extracellular vesicle miRNAs in therapeutic resistance are explained in Section 7.

**Table 1 cells-09-00029-t001:** Drug transporter-related miRNAs and their effects on the susceptibility of cancer cells to anti-cancer treatments.

MiRNAs	Target Gene(s)	Cancer Type	Effect of MiRNAs	Ref.
miR-20a-5p	MAPK1	Breast cancer	Over-expression of miR-20a-5p increases the overall cytotoxicity of several agents, such as vinorelbine, doxorubicin, and paclitaxel	[44]
miR-27-3p	FZD7	Hepatocellular carcinoma	Over-expression of miR-27-3p enhances the sensitivity of multidrug-resistant cells to 5-fluorouracil	[33]
miR-122-5p	CTNNB1	Hepatocellular carcinoma	Up-regulation of miR-122-5p raises the anti-cancer effect of oxaliplatin	[34]
miR-129-5p	ABCB1	Gastric cancer	Up-regulation of miR-129-5p heightens cisplatin-induced cell death and caspase activation	[17]
miR-133a-3p	ABCC1	Hepatocellular carcinoma	Over-expression of miR-133a-3p leads to elevated cytotoxicity of doxorubicin	[23]
miR-148a-3p miR-148b-3p miR-152-3p	SPIN1	Breast cancer	Over-expression of these miRNAs re-sensitizes the drug-resistant cells to doxorubicin	[45]
miR-199a-3p	MTOR	Cholangiocarcinoma	Reconstitution of miR-199a-3p increases growth inhibition rate and apoptosis induced by cisplatin	[39]
miR-210-3p	ABCC5	Pancreatic cancer	Elevated miR-210-3p levels improve the overall cytotoxicity of gemcitabine	[26]
miR-218-5p	PRKCE	Gallbladder cancer	Elevated miR-218-5p levels potentiate gemcitabine-mediated cell death and growth inhibition	[38]
miR-223-3p	ABCB1	Hepatocellular carcinoma	Down-regulation of miR-223-3p confers resistance to doxorubicin	[18]
miR-326	ABCC1	Hepatocellular carcinoma	Over-expression of miR-326 leads to elevated cytotoxicity of doxorubicin	[23]
miR-328-3p	ABCG2	Breast cancer	Over-expression of miR-328-3p augments the sensitivity of drug-resistant cells to mitoxantrone	[29]
miR-491-3p	ABCB1, SP3	Hepatocellular carcinoma	Down-regulation of miR-491-3p decreases the sensitivity to doxorubicin and vinblastin	[19]
miR-495-3p	ABCB1, UBE2C	Ovarian cancer, Gastric cancer, Lung cancer	Up-regulation of miR-495-3p re-sensitizes drug-resistant ovarian and gastric cancer cells to doxorubicin/paclitaxel combination, and cisplatin resistance is reversed in miR-495-3p over-expressing lung cancer cells	[20,46]
miR-506-3p	CTNNB1	Colorectal cancer	Over-expression of miR-506-3p re-sensitizes drug-resistant cells to oxaliplatin	[35]
miR-508-5p	ABCB1, ZNRD1	Gastric cancer	Down-regulation of miR-508-5p confers resistance to cisplatin, doxorubicin, vincristine, and 5-fluorouracil	[21]
miR-514	ABCA1, ABCA10, ABCF2	Ovarian cancer	Up-regulation of miR-514 re-sensitizes drug-resistant cells to cisplatin	[28]
miR-595	SLC19A1, ABCB1	Acute lymphoblastic leukemia, Ovarian cancer	Over-expression of miR-595 can either decrease or increase the efficacy of methotrexate or cisplatin, respectively	[30,31]
miR-1268a	ABCC1	Glioblastoma	Over-expression of miR-1268a augments temozolomide sensitivity	[24]

**Table 2 cells-09-00029-t002:** DNA damage repair-related miRNAs and their effects on the susceptibility of cancer cells to anti-cancer treatments.

MiRNAs	Target Gene(s)	Cancer Type	Effect of MiRNAs	Ref.
miR-7-5p	PARP1	Lung cancer	Over-expression of miR-7-5p increases the overall cytotoxicity of doxorubicin	[55]
miR-30-5p	FANCF, REV1	Breast cancer	Over-expression of miR-30-5p raises the anti-cancer effect of doxorubicin	[56]
miR-138-5p	ERCC1, ERCC4	Gastric cancer	Knockdown of miR-138-5p lowers the efficacy of cisplatin, thus enhancing cisplatin resistance	[58]
miR-182-5p	BRCA1, RAD51	Breast cancer, Acute Myelogenous Leukemia	Silencing of miR-182-5p results in resistance to PARP1 inhibitors and CNDAC	[63,64]
miR-205-5p	PRKCE, ZEB1	Prostate cancer	Reconstitution of miR-205-5p escalates the efficiency of radiotherapy	[68]
miR-211-5p	POLH, TDP1, ATRX, MRPS11, ERCC6L2	Ovarian cancer	Elevated miR-211-5p levels improve the overall cytotoxicity of carboplatin	[69]
miR-488-3p	EIF3A	Lung cancer	Elevated miR-488-3p levels impede cisplatin-mediated induction of cell death and growth inhibition	[73]
miR-493-5p	CHD4	Ovarian cancer	Down-regulation of miR-493-5p levels leads to enhanced responsiveness to cisplatin and olaparib	[75]
miR-520g-3p, miR-520h	APEX1	Multiple myeloma	Over-expression of both miR-520g-3p and miR-520h hampers the growth of bortezomib resistant multiple myeloma cells	[69]
miR-4429	RAD51	Cervical cancer	Over-expression of miR-4429 enhances radiosensitivity	[65]

**Table 3 cells-09-00029-t003:** Autophagy-regulating miRNAs and their effects on anti-cancer treatments.

MiRNAs	Target Gene(s)	Cancer Type	Effect of MiRNAs	Ref.
miR-23-3p	ATG12, HMGB2	Gastric cancer	Over-expression of miR-23-3p leads to the enhanced efficacy of 5-fluorouracil, cisplatin, and vincristine in drug-resistant cells	[90]
miR-26-5p	ULK1	Hepatocellular carcinoma	Over-expression of miR-26-5p promotes doxorubicin-induced apoptosis	[91]
miR-34-5p	HMGB1	Retinoblastoma	Reconstitution of miR-34-5p enhances cell death following treatment of etoposide, vincristine, and carboplatin	[92]
miR-101-3p	RAB5A, STMN1, ATG4D	Hepatocellular carcinoma, Breast cancer	Up-regulation of miR-101-3p increases cisplatin- and 4-hydroxytamoxifen-induced cell death in hepatocellular carcinoma and breast cancer cells, respectively	[93,94]
miR-129-3p	MTOR	Hepatocellular carcinoma, Gastric cancer	Silencing of miR-129-3p escalates the efficiency of Trichostatin A	[87]
miR-137-3p	ATG5	Pancreatic cancer	Elevated miR-137-3p levels enhances the effects of doxorubicin on growth inhibition and apoptosis	[95]
miR-140-5p	HMGN5	Osteosarcoma	Reconstitution of miR-140-5p sensitizes cells to cisplatin, doxorubicin, and methotrexate	[96]
miR-142-3p	ATG5, ATG16L1	Hepatocellular carcinoma	Reconstitution of miR-142-3p enhances the cytotoxicity of sorafenib	[97]
miR-148-3p	RAB12	Gastric cancer	Up-regulation of miR-148-3p reverses cisplatin resistance	[89]
miR-152-3p	ATG14	Ovarian cancer	Over-expression of miR-152-3p sensitizes cisplatin-resistant cells toward cisplatin via enhancing cell death and inhibiting cell growth	[98]
miR-214-3p	ATG12	Colorectal cancer	Down-regulation of miR-214-3p induces radioresistance	[99]
miR-224-3p	ATG5	Glioblastoma, Astrocytoma	Over-expression of miR-224-3p enhances the efficacy of temozolomide with increased apoptosis induction	[100]
miR-409-3p	BECLIN1	Colorectal cancer	Replacement of miR-409-3p sensitizes resistant cancer cells to oxaliplatin	[101]
miR-410-3p	HMGB1	Pancreatic cancer	Over-expression of miR-410-3p improves gemcitabine-induced cell death and growth inhibition in drug-resistant cells	[102]
miR-520-3p	ATG7	Hepatocellular carcinoma	Replacement of miR-520-3p increases the sensitivity of drug-resistant cells to doxorubicin by enhancing cell death and growth inhibition	[103]
miR-874-3p	ATG16L1	Gastric cancer	Restoration of miR-874-3p sensitizes cells to 5-fluorouracil and cisplatin	[104]

**Table 4 cells-09-00029-t004:** Stemness-regulating miRNAs and their effects on anti-cancer treatments.

MiRNAs	Target Gene(s)	Cancer Type	Effect of MiRNAs	Ref.
miR-7-5p	YY1	Glioblastoma	Treatment with miR-7-5p enhances the sensitivity of drug-resistant cells to temozolomide	[148]
miR-34-5p	NOTCH1, GOLPH3	Breast cancer, Urothelial bladder cancer	Ectopic expression of miR-34-5p increases the sensitivity to doxorubicin by enhancing apoptosis induction in drug-resistant breast cancer cells. Down-regulation of miR-34-5p desensitizes bladder cancer cells to gemcitabine and cisplatin	[122,147]
miR-93-3p, miR-105-5p	SFRP1	Breast cancer	Silencing of miR-93-3p and miR-105-5p enhances the sensitivity to cisplatin and chemoradiotherapy	[116]
miR-124-3p	USP14	Lung cancer	Over-expression of miR-124-3p increases the effects of gefitinib on apoptosis and growth inhibition	[120]
miR-136-5p	NOTCH3	Ovarian cancer	Over-expression of miR-136-5p escalates paclitaxel-induced cell death in drug-resistant cells	[124]
miR-139-5p	NOTCH1	Colorectal cancer	Ectopic expression of miR-139-5p enhances the sensitivity of CD44+/CD133+ cells to oxaliplatin, vincristine, 5-fluorouracil, and mitomycin C	[123]
miR-145-5p	c-MYC, CD44, KLF4, OCT4	Colorectal cancer, Gastric cancer	Over-expression of miR-145-5p sensitizes cells to cisplatin and 5-fluorouracil in gastric cancer. This miRNA also enhances the efficacy of radiation and oxaliplatin	[144,145]
miR-186-5p	YY1	Glioblastoma	Ectopic expression of miR-186-5p improves the cisplatin cytotoxicity	[149]
miR-195-5p	NOTCH2, RBPJ	Colorectal cancer	Over-expression of miR-195-5p subdues resistance to 5-fluorouracil	[126]
miR-196-5p	SOCS1, SOCS3	Colorectal cancer	Knockdown of miR-196-5p sensitizes cancer cells to 5-fluorouracil by augmenting apoptosis	[129]
miR-324-5p	SMO, GLI1	Multiple myeloma	Over-expression of miR-324-5p heightens the efficacy of bortezomib in multiple myeloma cells	[135]
miR-381-3p	NEFL	Glioblastoma	Silencing of miR-381-3p increases the sensitivity of cells to temozolomide	[150]
miR-423-5p	ING4	Glioblastoma	Over-expression of miR-423-5p significantly attenuates the chemosensitivity of glioma cells to temozolomide	[137]
miR-450b-5p	SOX2	Colorectal cancer	Ectopic expression of miR-450b-5p sensitizes cells to 5-fluorouracil	[143]
miR-589-5p	SOCS2, SOCS5, PTPN1, PTPN11	Hepatocellular carcinoma	Ectopic expression of miR-589-5p promotes the emergence of acquired resistance to doxorubicin	[131]
miR-873-5p	PD-L1	Breast cancer	Over-expression of miR-873-5p attenuates therapeutic resistance to doxorubicin	[142]
miR-1246	CCNG2	Oral cancer	Knockdown of miR-1246 sensitizes cancer cells to cisplatin	[117]

**Table 5 cells-09-00029-t005:** EMT-associated miRNAs and their effects on the susceptibility of cancer cells to anti-cancer treatments.

MiRNAs	Target Gene(s)	Cancer Type	Effect of MiRNAs	Ref.
miR-1-3p	MET	Lung cancer	Over-expression of miR-1-3p increases the anti-proliferative effects of gefitinib	[158]
miR-17-5p	DEDD	Gastric cancer	Inhibition of miR-17-5p augments cisplatin- and 5-fluorouracil-induced apoptosis	[161]
miR-103-3p	PRKCE	Lung cancer	Enforced expression of miR-103-3p elevates the anti-proliferative effects of gefitinib along with caspase 3/7 activation	[159]
miR-128-3p	ZEB1	Prostate cancer	Over-expression of miR-128-3p improves the effect of cisplatin on cell growth and invasion	[162]
miR-200 family	ZEB1, ZEB2	Gastric cancer, Breast cancer	Enforced expression of miR-200 family restores trastuzumab and cyclophosphamide sensitivity in gastric and breast cancer, respectively	[163,164]
miR-203a-3p	SRC	Lung cancer	Enforced expression of miR-203a-3p elevates the anti-proliferative effects of gefitinib along with caspase 3/7 activation	[159]
miR-204-5p	TGFBR2, ZEB1	Gastric cancer, Prostate cancer	Over-expression of miR-204-5p improves the efficacy of 5-fluorouracil in gastric cancer cells. In prostate cancer cells, miR-204-5p promotes docetaxel-mediated apoptosis	[154,165]
miR-206	MET	Lung cancer	Over-expression of miR-206 increases the anti-proliferative effects of gefitinib	[158]
miR-363-3p	SNAI1	Ovarian cancer	Silencing of miR-363-3p diminishes the anti-proliferative effects of cisplatin	[166]
miR-483-3p	ITGB3	Lung cancer	Epigenetic silencing of miR-483-3p desensitizes cells to gefitinib	[154]
miR-509-5p	VIM, HMGA2	Pancreatic cancer	Over-expression of miR-509-5p increases the anti-proliferative effects of gemcitabine	[153]
miR-574-3p	ZEB1	Gastric cancer	Enforced expression of miR-574-3p elevates cisplatin-induced apoptosis	[167]
miR-708-3p	ZEB1, CDH2, VIM	Breast cancer	Over-expression of miR-708-3p augments doxorubicin-mediated apoptosis	[168]
miR-873-5p	ZEB1	Breast cancer	Ectopic expression of miR-873-5p elevates the gemcitabine-induced cell growth arrest	[169]
miR-1243	SMAD2, SMAD4	Pancreatic cancer	Over-expression of miR-1243 increases the anti-proliferative effects of gemcitabine	[153]

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
