# Peer review of "MicroRNA-Based Combinatorial Cancer Therapy: Effects of MicroRNAs on the Efficacy of Anti-Cancer Therapies"

_cells, 2019, doi:10.3390/cells9010029_

Round 1
Reviewer 1 Report
The authors around Hyun Ah Seo et al. give a review on
an established topic: the acquired resistance of cancer cells
against established therapeutic regiments.
Here they focus specifically on the role of microRNAs in this
scenario and how a control of microRNA levels might be a key
to reverse the cellular resistance.
Obviously, it is established knowledge that clonal selection
might be a core mechanism in this process. Because every microRNA
plays a certain role in every cell, targeting the right cells
is still the primary challenge in all more specific therapeutic
approaches. This remains due even if they follow an adjuvant
concept.
The topic is not the bleeding edge of medical research but relevant.
Some incomprehensibility and conceptual flaws need to be corrected-
see Details.
Overall, the authors are not delivering the quality of the authors
they cite. Improving argumentation and discourse is mandatory.
Details
Keywords: sub-optimal collection in an random order
Title
The title is not really well formed English and needs revision.
Abstract
21-22
... This beneficial capacity of miRNAs stems
from their influences on the causes of therapeutic resistance. In..
From here on to the end of the abstract the conceptual line-up
needs more stringency and clarity. The final objectives are nebulous.
Please rewrite completely.
Introduction
33-39
Why starting with an unspecified enumeration?
resistance-mediating causes : not causes instead something like phenotypes
This section needs to be replaced by a thorough introduction to the
basic scenario on a conceptional and molecular level.
45-50
What does this section adds to the gentle reader's knowledge in terms
of understanding the term 'therapeutic resistance' and microRNA
action? Close to nothing.
The last sentence is an unproved statement.
--insufficient and logically cumbersome.
51-57
--is repeating the previous paragraph but not adding new insight
Overall the introduction needs much more stringency and a conceptual,
step-wise structure- less name dropping, more explanation/argumentation.
60-175
Figure 1 seems to be merging all information of the paragraph, but
is not specifying different cancer entities / cell lines-
would be interesting to code this into the Figure or
making a table.
The color coding of Figure 1 needs an explanation in the legend
or should be removed.
To many review citations.
If review citations, then port also the concepts in these reviews.
The structure resembles an enumeration but conceptualization of these
information is missing.
176-225
What should the reader learn from this lengthy section?
Table 1 is fine.
228-322
Figure 2 -> color coding?
If all details are in Figure 2, make a table for associating factors
with citations and focus in the writing on concepts.
--and so on...
One further issue is:
The sections are following a concept - but which one?
Here some guidance needs to be drafted and given in the introduction.
Overall the text needs to be shorted and tuned towards concepts.
The authors sink into their information.
536-559 Conclusion
545 our study -> our review
The conclusion is not on the provided huge amount of information instead
partly on new issues and aspects - revise.
Some typos and language issues here and there.
Reviewer 2 Report
I find the review by Ah Seo et al. a mere list of miRNAs involved in cancer resistance to therapy. The review could have been better structured in order to capture the attention of the general reader. However, it can be useful to people specifically working in the field of cancer resistance to therapy. If the Editor believe that that was the main goal of the review, the work is adequate. English language and style needs to be edited because some sentences sound really awkward. Here are some examples among many. Line 36: resistance include cancer stem cells (suggestion: resistance can be related to cancer stem cells). Line 48: decoding processes (mandatory change: translation). Line 48: augmentation (suggestion: induction). Line 250: treatment of paclitaxel (mandatory change: with paclitaxel). Line 261: converted by p53 (mandatory change: dependent on p53). Instead of using the expression "the resistance of cancer" one must use the expression "cancer resistance to therapy".
In the introduction on EMT the main pathway involved in EMT modulation, TGF beta family, must be quoted.
Reviewer 3 Report
The manuscript by Seo and colleagues entitled "The potential of MicroRNA modulation as therapeutic strategies for prevailing over the resistance of cancer" really does not reveal the review I thought it would. There is an enormous amount of author time spend reviewing the reviews of others to detail signaling cascades, survival pathways, and drug transport mechanisms that are involved in the resistance to therapy in a wide range of cancers. About half of the manuscript actually outlines where microRNAs may have a role in resistance to therapies. The autophagy section, in particular, is quite lengthy with transient attention to the role miRNAs might play in resistance. The transporter section is by far the best in describing how microRNAs modulate therapeutic outcomes. The Figures are complex and tedious to follow. The authors pay little, if any, attention to the "other-target" effects of the microRNAs they choose to highlight, with the exception of the mir-200c molecule. These small RNAs may have many targets including the ones noted.
Specifically, I would re-title Table 1 to be consistent with the other table titles. Line 274 refers to the "chemosensitization abilities" of microRNAs. These are outcomes observed with expression of these RNAs, not a choice job of the molecule. Lines 522-525 would be better placed in the 3rd paragraph of Section 6. The second sentence in section 2 is nearly the same as the first sentence in the second paragraph of section 2. Attention should be paid to the references as many gene names are not capitalized. Why is there no table of microRNAs/targets/cancers for Sections 3-5?
Round 2
Reviewer 1 Report
Overall the authors improved the manuscript a lot
and are now more conceptional.
Still an explanation of the color coding in,
especially the first two Figures, is missing and
needs to be integrated into the Figure legends.
E.g. green, light green, blue, red, sand,...
What does this grouping of the molecular factors means?
The manuscript might be now on an acceptable level.
Author Response
Comments and Suggestions for Authors
Overall the authors improved the manuscript a lot and are now more conceptional. Still an explanation of the color coding in, especially the first two Figures, is missing and needs to be integrated into the Figure legends. E.g. green, light green, blue, red, sand,...What does this grouping of the molecular factors means? The manuscript might be now on an acceptable level.Response 1: We greatly appreciate about comments and suggestions. Based on Reviewer 1’s suggestions, we have added the explanations about the color coding and arrows/lines in the Figure legends, as shown below.
1) Figure 1 (Page 4) - Rounded rectangles indicate miRNAs (light green), transcription factors (orange), cytoplasmic signaling molecules (light blue), and a transmembrane receptor (red). Activation is denoted by solid line arrows, and inhibitory effects are indicated by perpendicular lines. Dashed arrows represent the nuclear translocation of transcription factors.
2) Figure 2 (Page 16) - Rounded rectangles denote miRNAs (light green), stemness factors (orange), and upstream regulators of stemness factors (light blue, red, and green). Inhibitory effects are indicated by perpendicular lines. The positive regulation of stemness factors by each upstream factor is represented by solid lines.
3) Figure 3 (Page 19) - Rounded rectangles represent miRNAs (light green), miRNA targets (orange), and signaling factors/cellular events affected by miRNA targets (light orange). Activation is indicated by solid line arrows, and inhibitory effects are demonstrated by perpendicular lines. The secretion of extracellular vesicles is denoted by dashed arrows.
Reviewer 3 Report
The revised manuscript by Seo and colleagues is much improved. I appreciate the attention to the critiques. I have a few nit-picky comments remaining. The first sentence of the introduction states that "although cancer cells initially respond to treatments....". I would rather see phrases like this include a "may" or "often cancer cells respond to treatments". In fact, many tumors, and cell lines, are inherently resistance to therapy. The conclusions accurately outlines the complexities of resistance.
Overall, I think every new section should start with an introductory sentence. Several seem to run over from the previous section. For instance,3.2.4 to 3.2.5 or 3.3.1 to 3.3.2. That is just personal preference.
The conclusion is also much better. The appropriate level of caution was placed on the possibility for positive and negative outcomes from a single miR.
Author Response
Comments and Suggestions for Authors
The revised manuscript by Seo and colleagues is much improved. I appreciate the attention to the critiques. I have a few nit-picky comments remaining. The first sentence of the introduction states that "although cancer cells initially respond to treatments....". I would rather see phrases like this include a "may" or "often cancer cells respond to treatments". In fact, many tumors, and cell lines, are inherently resistance to therapy.Response 1: We agree with the Reviewer and have included “may” to the sentence (Page 1).
The conclusions accurately outlines the complexities of resistance. Overall, I think every new section should start with an introductory sentence. Several seem to run over from the previous section. For instance,3.2.4 to 3.2.5 or 3.3.1 to 3.3.2. That is just personal preference. The conclusion is also much better. The appropriate level of caution was placed on the possibility for positive and negative outcomes from a single miR.Response 2: We greatly appreciate about comments and suggestions. Based on Reviewer 3’s suggestions, we have revised and presented introductory sentences in Section 3.2., 3.2.3., 3.2.4., 3.2.5., 3.3., 3.3.1., and 3.3.2 (Pages 6-8), as shown below.
1) Section 3.2 - Several miRNAs have been identified to suppress cellular factors associated with DNA repair pathways, implying a possibility that the over-expression of a miRNA acting as the repressor of DNA repair could have a therapeutic benefit in cancer.
2) Section 3.2.3 - The excision repair cross-complementation group (ERCC) is known to participate in the nucleotide excision repair (NER) pathway [57].
3) Section 3.2.4 - BRCA1 DNA repair associated (BRCA1) functions in the repair of DNA double-stranded breaks by enhancing the recombinase activity of RAD51 recombinase (RAD51) [62].
4) Section 3.2.5 - ZEB1 has been proven to be critical for the regulation of checkpoint kinase 1, which coordinates DNA damage response signaling. Also, PKCε could activate DNA-dependent protein kinase by modulating the nuclear accumulation of epidermal growth factor receptor [67].
5) Section 3.3 - By contrast, miRNAs are capable of promoting DNA damage repair and stabilizing DNA replication fork, resulting in the aggravation of therapeutic resistance. It indicates that the knockdown of a resistance-associated miRNA could have a therapeutic benefit in cancer.
6) Section 3.3.1 - Eukaryotic translation initiation factor 3 subunit A (EIF3A) has been demonstrated to down-regulate NER activity by regulating the levels of NER factors. Therefore, the knockdown of EIF3A interrupts DNA damage-induced cell death [71,72].
7) Section 3.3.2 - Stabilization of DNA replication fork is one of the causes of therapeutic resistance. For example, the depletion of a chromatin-remodeling factor, such as chromodomain helicase DNA binding protein 4 (CHD4), confers cisplatin resistance in BRCA2-mutated cancer cells [74].